# Retinal Injury Activates Complement Expression in Müller Cells Leading to Neuroinflammation and Photoreceptor Cell Death

**DOI:** 10.3390/cells12131754

**Published:** 2023-06-30

**Authors:** Steven J. Tabor, Kentaro Yuda, Jonathan Deck, Gopalan Gnanaguru, Kip M. Connor

**Affiliations:** 1Angiogenesis Laboratory, Department of Ophthalmology, Massachusetts Eye and Ear Infirmary, Harvard Medical School, Boston, MA 02114, USA; 2Tulane University School of Medicine, Tulane Medical Center, New Orleans, LA 70112, USA

**Keywords:** neuroinflammation, neurodegeneration, microglia, complement, Müller cells, photoreceptor loss

## Abstract

Retinal detachment (RD) is a neurodegenerative blinding disease caused by plethora of clinical conditions. RD is characterized by the physical separation of retina from the underlying retinal pigment epithelium (RPE), eventually leading to photoreceptor cell death, inflammation, and vision loss. Albeit the activation of complement plays a critical role in the pathogenesis of RD, the retinal cellular source for complement production remains elusive. Here, using C3 tdTomato reporter mice we show that retinal injury upregulates C3 expression, specifically in Müller cells. Activation of the complement cascade results in the generation of proinflammatory cleaved products, C3a and C5a, that bind C3aR and C5aR1, respectively. Our flow cytometry data show that retinal injury significantly upregulated C3aR and C5aR1 in microglia and resulted in the infiltration of peripheral immune cells. Loss of C3, C5, C3aR or C5aR1 reduced photoreceptor cell death and infiltration of microglia and peripheral immune cells into the sub-retinal space. These results indicate that C3/C3aR and C5/C5aR1 play a crucial role in eliciting photoreceptor degeneration and inflammatory responses in RD.

## 1. Introduction

Retinal detachment (RD) is neurodegenerative blinding condition caused by a wide variety of retinal pathologies such as trauma, rhegmatogenous RD, neovascular age-related macular degeneration, and diabetic retinopathy [1,2,3]. In these clinical conditions, the physical separation of retina from the retinal pigment epithelium (RPE) leads to irreversible photoreceptor loss, as early as 12 h [4]. The activation of the photoreceptor cell death pathway can be mediated through caspase dependent or independent pathways [5,6]. Although the exact mechanism of initiation of photoreceptor cell death is not fully understood in RD, activation of the alternative complement pathway plays a major role in eliciting photoreceptor loss [2,4,7]. 

The complement system plays a major role in immunosurveillance and the host-defense mechanism [8]. The complement cascade can be activated through lectin, alternative, or classical pathways, all of which will lead to the activation of the central complement component 3 (C3) [8,9]. The activation and processing of C3 results in the release of C3a and C3b peptides; the C3b peptide facilitates the further cleavage of C5 into C5a and C5b fragments. The release of C3a and C5a fragments has been shown to be a potent influencer of inflammatory response [10,11,12,13]. Downstream activation of complement components can trigger the formation of the C5b-C9 membrane attack complex, which, once bound, can disrupt cell membrane integrity and contribute to cellular lysis [8,14]. Collectively, unregulated complement activation can directly impact photoreceptor cell death through selective clearance mechanisms, as well as promoting a pro-inflammatory local environment, both detrimental to long-term survival of photoreceptors.

C3 can be activated via multiple complement pathways, however, alternative pathway-dependent C3 activation plays a predominant role in RD pathobiology [4]. Inhibition or deficiency of C3, or the loss of factor B (a central activator of alternative pathway), significantly reduces RD-induced photoreceptor loss [4]. Despite knowing that C3 is a key player in RD pathobiology, the retinal source for C3 production in RD remains unclear. In this study, we explored the retinal cellular source for C3 production and receptor activation during RD-induced photoreceptor loss.

Our data show that the deficiency of C3 and C5 significantly suppressed RD-induced photoreceptor cell death and sub-retinal immune cell infiltration. C3 tdTomato reporter mice revealed that C3 expression was localized to Müller cells 24 h following RD injury. Flow cytometric studies showed that C3a binding C3aR and C5a binding C5aR1 expression levels were dramatically increased in microglia with RD. Loss of C3aR and C5aR1 significantly mitigated RD-induced photoreceptor cell death and sub-retinal immune cell infiltration into the injured area, suggesting that C3a/C3aR and C5a/C5aR1 activation contribute to RD-induced photoreceptor loss and inflammatory response. 

## 2. Materials and Methods

### 2.1. Animals

C57BL/6J (stock no. 000664), C3 knockout (KO) (stock no. 029661), C5 KO (stock no. 000461), C3aR KO (stock no. 033904) breeding pairs were purchased from Jackson Laboratories and maintained on a C57BL/6J background. C5aR1 KO mice on a C57BL/6J background were a gift from Dr. John Lambris (Perelman School of Medicine, University of Pennsylvania). All the mutant strains were crossed with C57BL/6J to create wild-type (WT) and KO lines and were used for the experiments. Floxed C3 IRES tdtomato knock-in reporter mice were obtained from Dr. Kemper (Immunology Center, National Heart, Lung, and Blood Institute, NIH, Bethesda, MD, USA) and backcrossed six generations to C57BL/6J background [15]. Mice were housed in a temperature-controlled facility with a 12 h light/12 h dark cycle. All animal experiments were conducted with male mice between the ages of 6–8 weeks in compliance with the purposed study procedures of Massachusetts Eye and Ear Animal Care Committee (protocols 2021N000034 and 2021N000013). 

### 2.2. Retinal Detachment Mice Model

Induction of retinal detachment was performed as previously described [16]. In brief, 6–8-week mice were anesthetized with an intraperitoneal injection of 2,2,2-tribromoethanol (250 mg/kg; Sigma-Aldrich Corp, St. Louis, MO, USA). After confirmation with toe pinch, pupils were dilated with topical phenylephrine (5%) and tropicamide (0.5%, Sandoz, Princeton, NJ, USA) and topical anesthesia (0.5% proparacaine hydrochloride ophthalmic solution, Sandoz, Princeton, USA) was applied. A scleral tunnel was created, first by an incision in the temporal conjunctiva at the posterior limbus to expose the sclera following a sclerotomy with a 30-gauge needle with the bevel pointed up. To decrease intraocular pressure, the corneal was punctured parallel to the iris with a 30 g needle. Using a Hamilton syringe, 3.5 µL of sodium hyaluronate (Provisc^®^, Alcon, Fort Worth, TX, USA) was injected to detach the retina from the underlying RPE. Lastly, the self-sealing scleral incision was reinforced with surgical glue (Webglue: Patternson Veterinary, Loveland, CO, USA) and the conjunctiva was resecured at the posterior limbus. Detachments with sub-retinal hemorrhaging or incomplete detachments were excluded from all analysis. 

### 2.3. TUNEL Labeling and Quantitation of Photoreceptors in the Outer Nuclear Layer (ONL)

Eyes with retinal detachment were stained with terminal deoxynucleotidyl transferase–mediated deoxyuridine triphosphate nick end labeling (TUNEL) apoptosis kit as previously described [17]. Briefly, detached retinas were carefully enucleated and embedded in O.C.T compound (Tissue-Plus; Fisher Scientific, Waltham, MA, USA) with the detach regions facing down. Non-consecutive serial sections were captured at 10 µm thickness on a cryostat (CM1950; Lecia) at the peak of the detached region approximately 800–1000 µm from the self-sealing wound. Detached sections were stained with a TUNEL apoptosis kit following the manufacture’s protocol (EMD Millipore, Burlington, VT, USA) and stained with DAPI nuclear dye. The ONL layer of adjacent sides of the detached retina were imaged with an epifluorescence microscope (Axio Observer Z1; Zeiss Oberkochen, Germany) using a 40X lens. TUNEL positive cells were manually counted using the cell counter function on ImageJ for four sections per detached retina. The ONL thickness was measured by manually outlining using ImageJ freehand tool for at least eight regions per retina. The total representative TUNEL positive cells per mm^2^ was calculated by dividing the averaged TUNEL positive cells per averaged ONL area.

### 2.4. Immunostaining and Subretinal Immune Cell Counts 

Anesthetized mice were perfused with 1X PBS, and eyes were enucleated. Whole eyes were fixed for 15 min in 4% PFA in 2X PBS. After a brief wash, the lens and cornea were removed, and the posterior eyecups were fixed in 4% PFA In 2X PBS for an additional 30 min. Eyecups were then transferred to 1X PBS for 10 min and cryopreserved in a 10%, 20% and 30% sucrose gradient. Eyes were then embedded in O.C.T compound with the detached retina facing down. The eyecups were then sectioned at 20 µm and used for IHC. 

To label Müller cells, detached and undetached retinal sections of C3 tdTomato reporter mice were blocked in blocking buffer (0.3% Triton X-100, 0.2% BSA and 5% goat serum in 1X PBS) for 1 h at room temperature. Retinal sections were incubated, then incubated with rabbit anti-glutamate synthetase (1:100, cat no. GTX109121, Gentex, Irvine, USA). Sections were then washed 3 times in 1X PBS followed by 1 h incubation with secondary antibody (goat anti-rabbit AF488, cat no. A55053, ThermoFisher, Waltham, MA, USA) and DAPI nuclear stain for 10 min at room temperature. The sections were washed and then mounted with anti-fade mounting media (PermaFluor™ Aqueous Mounting Medium, ThermoFisher, Waltham, MA, USA). The specimens were imaged using Leica SP8 confocal microscope. 

To count infiltrated sub-retinal immune cells, retinal sections were blocked in blocking buffer (0.3% Triton X-100, 0.2% BSA and 5% goat serum in 1X PBS) for 1 h at room temperature. Subsequently, sections were incubated with primary antibodies: rabbit anti-P2ry12 (1:500, from Anaspec, Fremont, CA, USA) and AF488 rat anti-CD11b (1:100, Clone M1/70 from abcam, Waltham, MA, USA). Following primary overnight incubation, sections were washed 3 times in 1X PBS for 10 min and incubated with a secondary antibody goat anti-rabbit AF594 (1:500 cat. no A-11037, ThermoFisher, Waltham, MA, USA) for 1 h at room temperature. Next, sections were washed and stained with DAPI and then mounted with anti-fade mounting media (PermaFluor™ Aqueous Mounting Medium, ThermoFisher, Waltham, MA, USA). Whole subretinal regions were manually counted for P2ry12^+^ CD11b^+^ (microglia) and P2ry12^−^ CD11b^+^ (macrophage) cells for eight detached regions per retina using an epifluorescence microscope (Axio Observer Z1).

### 2.5. Flow Cytometric Analyzes of Complement Receptors

Retinal single cell suspensions were prepared as previously described [18,19]. In brief, anesthetized mice were perfused with 20 mL of ice-cold PBS and eyes were enucleated. Two retinas per animal were isolated and incubated in digestion buffer (HBSS supplemented with 10% FBS, 10 mM HEPES, 0.7 mg/mL calcium chloride (Sigma-Aldrich), 1 mg/mL collagenase D type II, and 0.1 mg/mL DNase I (Roche) for 45 min at 37 °C. After incubation, single cell suspensions were created by careful trituration with pipette and filtered with a 40 µm cell strained. To remove large debris, cell suspensions were centrifuged at 50× *g* for 1 min. The supernatant was collected, and centrifuged at 350× *g* at 4 °C. The resulting pellet was washed and resuspended in FACS buffer. Single cell suspensions were blocked with goat anti-CD16/32 (Clone 2 g.4 from Absolute Antibody, Boston, CA, USA) for 20 min at 4 °C. After blocking, the following primary antibodies were used: BV785 rat anti-mouse CD45 (Clone 30-F11 from Biolegend, San Diego, CA, USA), BB515 rat anti-CD11b (Clone M1/70, from BD biosciences, Franklin Lakes, NJ, USA), APC rat anti-mouse P2RY12 (Clone S16007D, from Biolegend), rat anti-mouse C3aR (Clone 14D4, from Hycult, Wayne, NJ, USA) conjugated with (cat. no ab102918 from abcam) APC Rat anti-mouse C5aR1 (Clone 20/70, from Biolegend, San Diego, CA, USA). Complement receptors C3aR and C5aR1 were evaluated on microglia identified as (CD45^mid^ CD11b^+^ P2ry12^+^) and macrophages (CD45^High^ CD11b^+^ P2ry12^−^).

### 2.6. Statistical Analysis

Experiment data are presented as the mean SEM. Statistical analysis between two groups were determined by unpaired students t-test after ROUT’s outlier test. Statistic significates was denoted as follows: * *p* < 0.05, ** *p* < 0.01, *** *p* < 0.001, and **** *p* < 0.0001.

## 3. Results

### 3.1. Müller Glial Cells Express C3 during Retinal Detachment, and Loss of C3 Protected Photoreceptor from Cell Death and Suppressed Sub-Retinal Immune Cell Infiltration

Upregulation of C3 plays a major role in RD pathobiology [4,7]. While retinal pigment epithelial cells are known to produce C3 [20], the retinal cellular source of C3 remains elusive. Here, we sought to determine the cell type(s) responsible for resident C3 production in the retina utilizing C3 tdTomato expressing reporter mice [15]. Following the induction of RD, animals were subsequently examined for C3 tdTomato expression at day 1 and day 3 post-acute injury. Analysis of C3 tdtomato reporter retinal cryosections stained for anti-glutamate synthase showed robust upregulation of C3 reporter expression in Müller cells as early as one day post-detachment (Figure 1A), which corresponds to the peak of photoreceptor cell death in this model [4,17]. Müller cells continued to express C3 even at day 3, post RD (Figure 1A). Importantly, Müller cell expression of C3 tdtomato without detachment was below the limits of detectability (Figure 1A). Our results suggest, that in response to retinal insult, Müller cells upregulate expression of C3. 

Since C3 expression was upregulated within 24 h post-RD, we next assessed photoreceptor cell death and sub-retinal immune cell infiltration in WT and C3 KO 24 h post-RD (peak of photoreceptor loss) [4] to further investigate the role of C3 in the pathogenesis of RD. Assessment of TUNEL reactivity in the ONL of detached retinas showed a significant decrease in photoreceptor cell death with C3 deficient mice compared to WT controls (Figure 1B). It has been suggested that microglia and macrophage cells, both professional phagocytes, play a role in photoreceptor cell death in RD, since these cells rapidly infiltrate the sub-retinal layer following retinal injury in RD [21]. Using a histological approach, we assessed microglia and macrophage localization into the sub-retinal space of detached WT and C3 KO mice with immunostaining of microglial specific marker P2ry12, and generic microglial/monocyte/macrophage marker CD11b. Examination of C3 deficient retinas displayed a reduced microglial and macrophage infiltration into the injured sub-retinal region compared to WT controls (Figure 1C). These results indicate a role for C3 in photoreceptor cell death and sub-retinal immune cell recruitment.
Figure 1Retinal detachment increased C3 expression in Müller glial cells and C3 loss suppresses photoreceptor cell death and sub-retinal immune cell infiltration. (**A**) Representative retinal cross sections of C3 tdtomato reporter mice without (control) and with retinal detachment (RD) at day 1 and day 3, immunostained for Müller cell specific marker glutamate synthase and DAPI nuclear stain revealing tdTomato expression in Müller cells (*n* = 3). (**B**) Representative WT and C3 KO retinal sections showing TUNEL reactivity of the outer nuclear layer (ONL) stained with DAPI nuclear stain. The bar graph shows the quantitation of TUNEL positive photoreceptor nuclei 24 h-post RD in WT (*n* = 7) and complement C3 KO (*n* = 6) retinal sections. (**C**) Representative detached WT (*n* = 7) and C3 KO (*n* = 6) retinal cross sections showing P2ry12, CD11b, and DAPI nuclear staining antibodies. White arrow; P2ry12^+^/CD11b^+^ microglia, red arrow; P2ry12^−^/CD11b^+^ macrophages. Bar graphs shows the quantitation of infiltrated subretinal P2ry12^+^/CD11b^+^ microglia and P2ry12^−^/CD11b^+^ macrophages in WT and C3 KO retinal cross sections. Statistical significance was analyzed by unpaired Student’s *t* test and all error bars represent mean S.E.M. * *p* ≤ 0.05, *** *p* ≤ 0.001; Scale bars (**A**) and (**B**) are 20 μm and (**C**) is 50 µm.
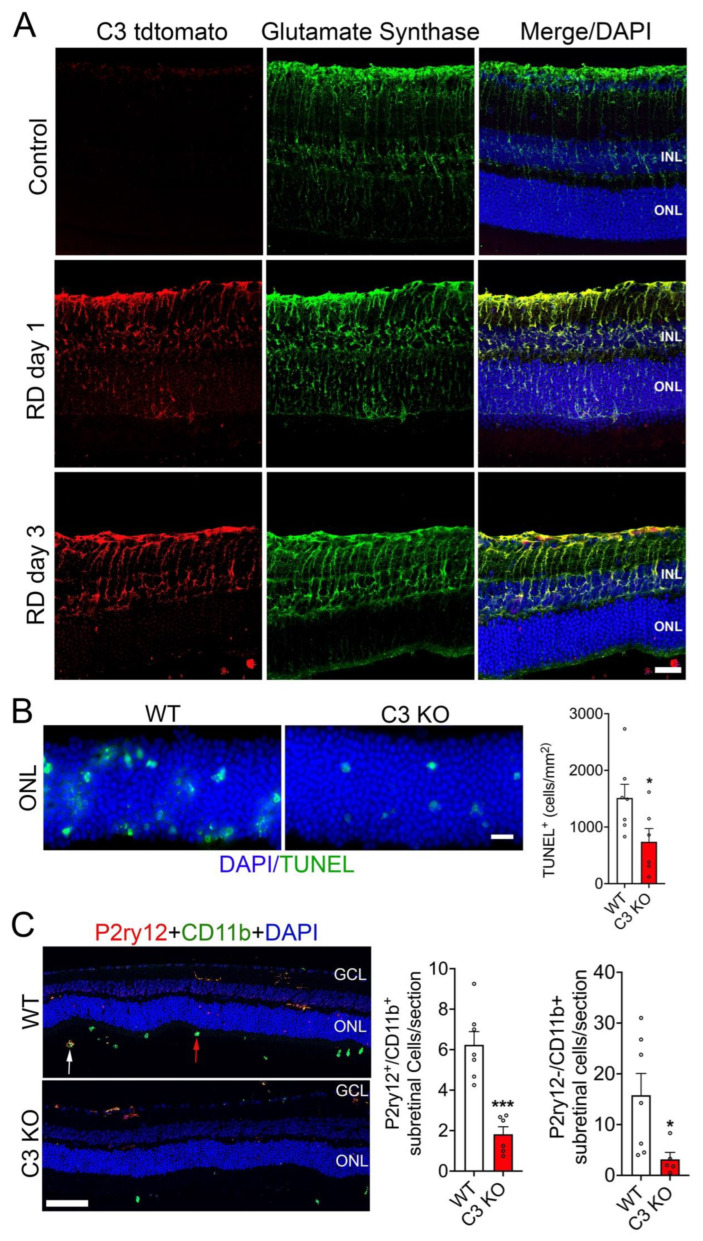



### 3.2. C3aR Is Upregulated in Microglia/Macrophages during RD, and Loss of C3aR Protected Photoreceptors from Cell Death and Suppressed Immune Cell Infiltration

C3a is generated through proteolytic cleavage of C3, leading to the induction of inflammatory responses [22,23]. However, the C3a-C3aR axis is ill-defined in response to RD. We first assessed C3aR expression in undetached and detached retinas of C57BL/6J mice using flow cytometric analysis to further identify the cell type(s) responsible for binding C3a in RD (Appendix A and Figure 2A). Retinas with and without detachment were stained with a C3aR specific antibody that showed approximately 99.6% of resident microglia (defined as CD45^mid^ CD11b^+^ P2ry12^+^) expressed C3aR in undetached retinas that was maintained at 99.7% post 24 h RD (Figure 2A). In addition, 22.2% of macrophage populations (defined as CD45^high^ CD11b^+^ P2ry12^−^) showed C3aR expression in undetached retinas, which are presumably circulating macrophage population. Moreover, the percentage of infiltrating macrophages with C3aR expression increased to approximately 79.5% post-24 h RD (Figure 2A). Analysis of mean fluorescent intensity (MFI) levels of C3aR in both microglia and macrophages demonstrated a significant upregulation post 24 h RD (Figure 2B,C). To determine if C3a/C3aR axis is involved in injury-induced photoreceptor cell death and sub-retinal inflammation, we next induced RD in C3aR KO and WT mice. Analysis of retinal cross sections of C3aR KO demonstrated decreased TUNEL reactivity in the ONL compared to WT (Figure 2D). Moreover, sub-retinal infiltration of P2ry12^+^/CD11b^+^ and P2ry12^−^/CD11b^+^ immune cells were significantly reduced in C3aR KO retinas compared to WT retinas (Figure 2E). These data illustrate that the C3a/C3aR axis plays a critical role in retinal injury-induced photoreceptor loss and the sub-retinal inflammatory response.

### 3.3. Loss of C5-Reduced Photoreceptor Cell Death and Suppressed Immune Cell Infiltration

Previous studies have identified a protective role in inhibiting the alternative pathway including complement Factor B and C3 [4]. However, the role of downstream complement protein C5 has yet to be fully identified. To interrogate the role of complement protein C5 in RD, we assessed the total number of TUNEL positive cells in WT and C5 KO retinas 24 h post retinal detachment. As we observed with the C3 deficient mice, there was a two-fold reduction in TUNEL positive cells in the ONL of C5 KO retinas compared to WT 24 h post RD (Figure 3A). In addition, immunohistochemistry analysis of detached WT and C5 KO retinal cross sections stained with P2ry12 (marker for microglia) and CD11b (marker for microglia/macrophages) were observed to have significantly reduced cell numbers in the subretinal space (Figure 3B). These results demonstrate that the loss of downstream components in the complement cascade, C5, preserves photoreceptor viability and decreases immune cell recruitment to the subretinal space in response to retinal detachment. 

### 3.4. C5aR1 Is Upregulated in Microglia/Macrophages during RD, and the Loss of C5aR1 Protected Photoreceptor from Cell Death and Suppressed Immune Cell Infiltration

Given that a deficiency in complement protein C5 has explicit roles in the neuroprotection of the retina in RD and its cleavage product, C5a, is a potent immune modulator [24]. We next sought to identify the immune cell(s) involved in the ligand binding of C5a with its cognate complement receptor C5aR1. To identify the C5aR1 expressing cell(s) in RD, we utilized flow cytometry to assess single cell suspensions from either undetached and detached retinas and stained for C5aR1 (Appendix A and Figure 4). Our results showed that approximately 95.1% of CD45^mid^ CD11b^+^ P2ry12^+^ retinal resident microglia were positive for C5aR1 expression and that 100% of resident microglia expressed C5aR1 in retinal detachment (Figure 4A). Furthermore, resident microglia of detached retinas showed a robust increase in C5aR1 expression (Figure 4B). Similarly, analysis of CD45^high^ CD11b^+^ P2ry12^−^ infiltrating macrophage showed 83.6% of the population expressing C5aR1 and that detached retinas increased C5aR1 expression when compared to undetached retinas (Figure 4A,C). 

We then sought to determine if the loss of C5aR1 reduces RD-induced photoreceptor cell death and sub-retinal immune cell infiltration. Analysis of C5aR1 KO retinal cross-sections displayed significantly reduced TUNEL positive cells in the ONL, compared to WT 24 h post-retinal detachment (Figure 4D). Further immunohistochemistry analysis of detached C5aR1 KO retinal cross-sections stained with P2ry12 (marker for microglia) and CD11b (marker for microglia/macrophages) were observed to have significantly reduced infiltration immune cell numbers in the subretinal space compared to WT (Figure 4E). These results illustrate that the C5/C5aR1 axis plays a critical role in eliciting immune cell infiltration and photoreceptor loss during retinal injury. 

## 4. Discussion

Complement activation plays a key role in eliciting an inflammatory response in several neurodegenerative conditions. In this study, we investigated the role of C3 and C5 activation in retinal injury-induced inflammatory response and photoceptor cell death. Our data show that retinal injury elicited C3 reporter expression specifically in Müller cells and that the loss of C3 or C3aR (C3a binding receptor) suppressed sub-retinal immune cell infiltration and photoreceptor cell death. Moreover, the loss of the downstream complement cascade component C5 and C5aR1 (C5a binding receptor) reduced sub-retinal immune cell infiltration and photoreceptor cell death. These results illustrate that the activation of the C3a/C3aR and C5a/C5aR1 axis plays a significant role in promoting inflammatory response and photoreceptor damage in response to retinal injury. 

Photoreceptors are high oxygen-consuming cells, and the choroid vascular supply fulfills the metabolic demand of photoreceptors through RPE cells [25]. In RD, the physical separation of photoreceptors from the RPE/choroid creates a hypoxic environment that accelerates photoreceptor cell death, in part, mediated through C3 activation [4,26,27]. During RD-induced photoreceptor damage, Müller cells undergo reactive gliosis [26,28]. Müller glial cells that span almost the entire thickness of the retina are key for maintaining retinal homeostasis and are important for visual cycle function [29,30]. Under pathological conditions, Müller cells undergo reactive gliosis and potentially contribute to inflammatory responses [29,31]. In RD, Müller cells are reported to undergo morphological changes and express the gliosis marker glial–fibrillary acidic protein [32]. However, it remains unclear if Müller cells reactivity contributes to RD pathobiology. Our study utilizing C3 reporter mice demonstrates that retinal injury induces C3 expression in Müller cells and that Müller cell derived C3 could potentially play a role in inducing photoreceptor damage and an inflammatory response. Such astroglial derived C3 activation of inflammatory response has been observed in the brains of neurodegenerative models [10]. During development, the C3 expression is predominantly observed in a subset of microglial population and presumably ganglion cells in the retina [18]. In contrast to developmental expression, in cases of retinal injury, our data show that C3 expression in the retina is localized to Müller cells. This illustrates that the retinal cellular source for C3 is context specific and that Müller cells are key the drivers of complement activation in response to acute retinal injury.

A recent finding suggests that intracellular C3a/C3aR-dependent downstream activation of the NLR family pyrin domain, containing three (NLRP3) inflammasome, plays a critical role in tissue priming leading to inflammation [11,33]. Although such intracellular activation of C3a/C3aR role in RD-induced tissue priming remains to be explored, reduction in photoreceptor cell death and sub-retinal immune cell infiltration in RD-induced C3 and C3aR null retinas further indicated a strong involvement of C3a/C3aR axis in accelerating tissue inflammation. 

In addition to C3a, the activation of complement cascade also generates bioactive C5a [34]. In human macrophages, C5a is shown to induce cytokine production, which promotes the inflammatory response [35]. The C5a excretes its pro-inflammatory function through binding and activation of C5aR1 receptor [36,37]. In neurodegenerative models, such as the Alzheimer’s, C5a/C5aR1 is suggested to promote disease progression [38]. An over expression of C5a has a profound effect on activating astrocytes and increasing microglial population in the brain [38]. Genetic ablation of C5aR1 delayed/decreased microglial population in an experimental model of Alzheimer’s disease [38]. Although we did not see any changes in microglial population following retinal injury, we observed a significant increase in peripheral macrophage infiltration into the retina. The reduction in sub-retinal microglial and macrophage infiltration in C5 and C5aR1 deficient mice suggests that C5a/C5aR1 also plays a key role in promoting RD pathogenesis. Taken together, our experimental model of RD suggests that the suppression of the C3a/C3aR and C5a/C5aR1 axis potentially represses the inflammatory response and contains photoreceptor damage in neurodegenerative conditions like RD. 

## 5. Conclusions

We have previously shown that complement activation plays a major role in neurodegenerative RD pathobiology. In this study, utilizing C3-tdTomato reporter mice, we have provided evidence that Müller glia is a key source for complement activation in response to retinal injury. Furthermore, our study shows that the loss of C3, C5, and their binding receptors, C3aR and C5aR1, significantly mitigated photoreceptor cell death and infiltration of immune cells into the sub-retinal space post injury. Although further studies are required, results from our study indicate that the inhibition of the C3/C3aR and/or C5/C5aR axis during the early phase of retinal injury could potentially prolong photoreceptor survival. 

## Figures and Tables

**Figure 2 cells-12-01754-f002:**
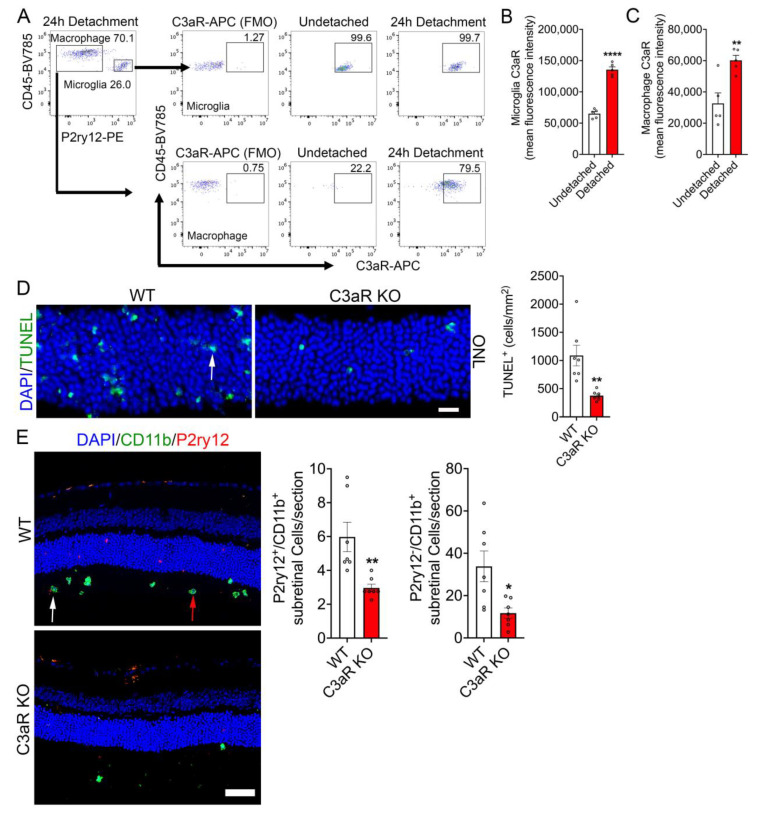
C3aR loss reduces retinal detachment induced photoreceptor loss and sub-retinal infiltration of immune cells. (**A**) Flow cytometry was performed on undetached (*n* = 5) and 24 h post detached retinas (*n* = 5) to assess C3aR expression in microglia and infiltrating macrophages. Microglia and macrophages were identified by gating on CD45^mid^ CD11b^+^ P2ry12^+^ and CD45^high^ CD11b^+^ P2ry12^−^, respectively, and further gated on C3aR^+^ cells to show microglia or macrophage specific C3aR expression. (**B**) Graphical representation of mean fluorescent intensity of C3aR expression in CD45^mid^ CD11b^+^ P2ry12^+^ microglia 24 h post retinal detachment. (**C**) Graphical representation of mean fluorescent intensity of C3aR expression in CD45^high^ CD11b^+^ P2ry12^−^ macrophage cell populations 24 h post retinal detachment. (**D**) Representative WT and C3aR KO retinal cross sections showing TUNEL reactivity (arrow) in the ONL co-stained with DAPI (*n* = 7). Bar graph is the quantitation of TUNEL positive photoreceptor nuclei 24 h-post retinal detachment. (**E**) Representative retinal cross sections showing P2ry12, CD11b, and DAPI nuclear staining in WT (*n* = 7) and C3aR deficient mice (*n* = 7). White arrow; P2ry12^+^/CD11b^+^ microglia, red arrow; P2ry12^−^/CD11b^+^ macrophages. Bar graphs shows the quantitation of infiltrated subretinal P2ry12^+^/CD11b^+^ microglia and P2ry12^−^/CD11b^+^ macrophages in WT and C3aR KO retinal cross sections. Statistical significance was analyzed by unpaired Student’s *t* test and all error bars represent mean S.E.M. * *p* ≤ 0.05, ** *p* ≤ 0.001, **** *p* ≤ 0.0001. Scale bar: (**D**) is 20 μm and (**E**) is 50 µm.

**Figure 3 cells-12-01754-f003:**
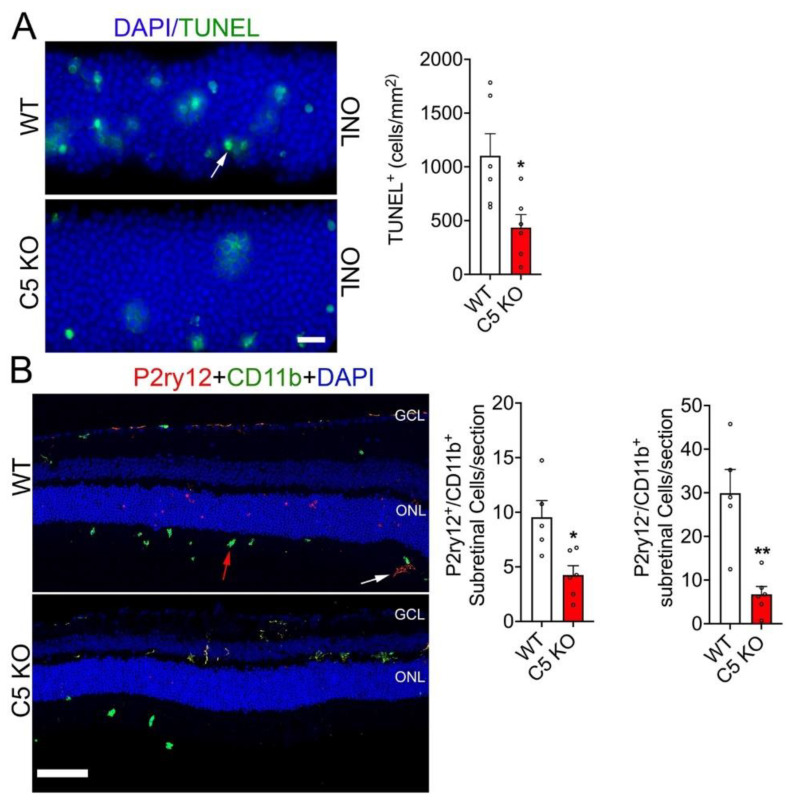
Loss of C5 decreased photoreceptor cell death and immune cell subretinal infiltration. Retinal sections 24 h post retinal detachment were assessed in WT (*n* = 6) and C5 KO (*n* = 6). (**A**) Representative TUNEL staining of the outer nuclear layer (ONL) counter-stained with DAPI and the bar graph is the quantitation of apoptotic photoreceptors 24 h-post retinal detachment of WT and complement C5 KO mice. (**B**) Representative immunohistochemistry of detached retinal cross sections stained with anti-P2ry12 (microglia) and anti-CD11b (macrophage/microglia) antibodies with DAPI in WT (*n* = 5) and C5 KO (*n* = 6). White arrow; P2ry12^+^/CD11b^+^ microglia, red arrow; P2ry12^−^/CD11b^+^ macrophages. Bar graph shows the quantitation of microglia (P2y12^+^CD11b^+^) and macrophage (P2y12^−^CD11b^+^) counts in the sub-retinal space of the whole detached region. Statistical significance was analyzed by unpaired Student’s *t* test and all errors bars represent mean S.E.M. * *p* ≤ 0.05, ** *p* ≤ 0.005; Scale bars: (**A**) is 20 μm and (**B**) is 50 µm.

**Figure 4 cells-12-01754-f004:**
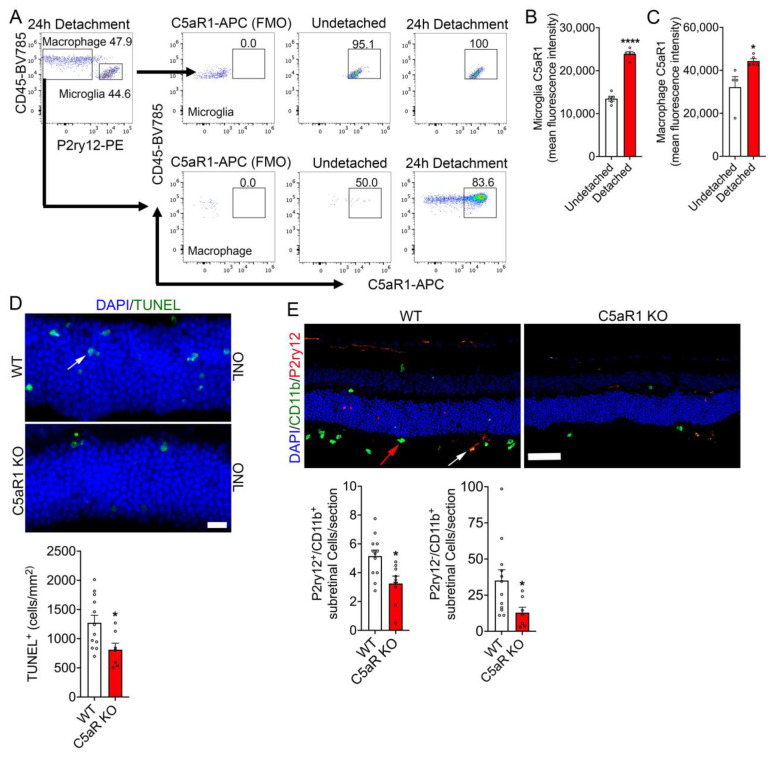
C5aR1 loss reduces retinal detachment-induced photoreceptor loss and sub-retinal infiltration of immune cells. (**A**) Flow cytometry was performed on undetached (*n* = 5) and 24 h post detached retinas (*n* = 5) to assess C5aR1 expression in microglia and infiltrated macrophages. Cell population gated for CD45^mid^ CD11b^+^ P2ry12^+^ C5aR1^+^ shows microglia specific C5aR1 expression and cell population gated for CD45^high^ CD11b^+^ P2ry12^−^ C5aR1^+^ shows macrophage specific C5aR1 expression. (**B**) Graphical representation of mean fluorescent intensity of C5aR1 expression in CD45^mid^ CD11b^+^ P2ry12^+^ microglia 24 h post retinal detachment. (**C**) Graphical representation of mean fluorescent intensity of C5aR1 expression in CD45^high^ CD11b^+^ P2ry12^−^ macrophage cell populations 24 h post retinal detachment. (**D**) Representative WT (*n* = 12) and C5aR1 (*n* = 7) retinal cross sections showing TUNEL reactivity in the ONL counter stained with DAPI. The bar graph is the quantitation of TUNEL positive photoreceptor nuclei 24 h post-retinal detachment. (**E**) RD-induced retinal cross sections of WT and C5aR1 were immunostained with anti-P2ry12 and anti-CD11b antibodies and DAPI nuclear stain. White arrow; P2ry12^+^/CD11b^+^ microglia, red arrow; P2ry12^−^/CD11b^+^ macrophages. Bar graphs are the quantitation of P2ry12^+^ CD11b^+^ microglia and P2ry12^−^ CD11b^+^ macrophage counts in the sub-retinal space. Statistical significance was analyzed by unpaired Student’s *t* test and all error bars represent mean S.E.M. * *p* ≤ 0.05, **** *p* ≤ 0.0001. Scale bar: (**D**) is 20 μm and (**E**) is 50 µm.

## Data Availability

All the data generated and analyzed during this study are provided in the manuscript and are available by reaching out to the lead authors.

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
