# Peer review of "Retinal Injury Activates Complement Expression in Müller Cells Leading to Neuroinflammation and Photoreceptor Cell Death"

_cells, 2023, doi:10.3390/cells12131754_

Round 1
Reviewer 1 Report
In this paper, Tabor and co-workers have studied the role of the central complement component (C3) after retinal detachment. Overall, the study is well designed, and the results are consistent among experiments. A few things I would to be addressed before accepting this paper for publication.
In most of their experiments using flow citometry, they used as a control, WT retinas without retinal detachment. However, this control is lost in their immunofluorescence experiments. If it is possible, it should be desirable to perform those experiments with WT retinas without retinal detachment.
There are other proteins whose role is very relevant during reactive gliosis in Müller cells, such as the GFAP, whose is expression is highly upregulated. I am wondering whether the expression of GFAP could be analyzed in those retinas.
Finally, I would like to know why the analysis were carried out 24 hours after the damage.
Author Response
We sincerely thank the reviewer for the valuable comments for the review of our manuscript “Retinal injury activates complement expression in Müller cells leading to neuroinflammation and photoreceptor cell death”. Please see below our point-by-point response addressing the concerns. We have also included a manuscript containing highlighted changes (major edits) as well as a cleaned version of the manuscript. We hope that you agree that it is now in a form appropriate for publication in ‘Cells’.
Reviewer 1
In this paper, Tabor and co-workers have studied the role of the central complement component (C3) after retinal detachment. Overall, the study is well designed, and the results are consistent among experiments. A few things I would to be addressed before accepting this paper for publication.
In most of their experiments using flow citometry, they used as a control, WT retinas without retinal detachment. However, this control is lost in their immunofluorescence experiments. If it is possible, it should be desirable to perform those experiments with WT retinas without retinal detachment.
The purpose of our flowcytometry experiment is to determine the expression level of complement receptors C3aR and C5aR1 in microglia and peripheral immune cells with and without detachment.
The purpose of immunofluorescence experiment is to capture the differences in the incidence of photoreceptor cell death and the infiltration of microglia and peripheral immune cells to the sub-retinal space in wild type and knockout retinas. Under normal undetached conditions, such events are absent. Shown below is an example of our ongoing study utilizing floxed-C3aR-tdTomato reporter mice revealing infiltration of microglia (CD11b+, C3aR+, and P2ry12+) and immune cells (CD11b+, C3aR+, and P2ry12-) into the sub-retinal space only with retinal detachment.
There are other proteins whose role is very relevant during reactive gliosis in Müller cells, such as the GFAP, whose is expression is highly upregulated. I am wondering whether the expression of GFAP could be analyzed in those retinas.
It has been previously reported that Müller cells undergo reactive gliosis in RD and express GFAP [1]. However, Müller cells involvement in complement upregulation has yet to be fully elucidated, until now. Thus, we chose a Müller cell specific marker to determine complement activation in the retina to avoid potential cross-reactivity of GFAP to astrocytes.
Finally, I would like to know why the analysis were carried out 24 hours after the damage.
Previously reported time course analysis in RD have identified 24 hrs post-RD as the apex of photoreceptor cell death and complement gene expressions including FactorB and C3 [2,3], which we have referred to in the manuscript. We therefore chose this time-point to determine the cellular source for C3 activation and investigated if the loss of complement activation subdues photoreceptor cell death and the infiltration of immune cells at an early phase.
References:
- Verardo, M.R.; Lewis, G.P.; Takeda, M.; Linberg, K.A.; Byun, J.; Luna, G.; Wilhelmsson, U.; Pekny, M.; Chen, D.F.; Fisher, S.K. Abnormal reactivity of muller cells after retinal detachment in mice deficient in GFAP and vimentin. Invest Ophthalmol Vis Sci 2008, 49, 3659-3665, doi:10.1167/iovs.07-1474.
- Matsumoto, H.; Kataoka, K.; Tsoka, P.; Connor, K.M.; Miller, J.W.; Vavvas, D.G. Strain difference in photoreceptor cell death after retinal detachment in mice. Invest Ophthalmol Vis Sci 2014, 55, 4165-4174, doi:10.1167/iovs.14-14238.
- Sweigard, J.H.; Matsumoto, H.; Smith, K.E.; Kim, L.A.; Paschalis, E.I.; Okonuki, Y.; Castillejos, A.; Kataoka, K.; Hasegawa, E.; Yanai, R.; et al. Inhibition of the alternative complement pathway preserves photoreceptors after retinal injury. Sci Transl Med 2015, 7, 297ra116, doi:10.1126/scitranslmed.aab1482.

Reviewer 2 Report
The paper entitle “Retinal injury activates complement expression in Müller cells leading to neuroinflammation and photoreceptor cell death” presented by Tabor SJ et al. is well-structured and results are very interesting for ophthalmology field. However this reviewer has some minor concerns that need to be address before publication.
The introduction although being brief is clear for people who has wealth of knowledge in the field, but if the authors wants to achieve more public they should explain briefly the main photoreceptor death pathways and the relevance of the complement in each one of them. Although this is not mandatory, it is only a suggestion.
In the methods section:
Authors should add the license number of the animal care committee
Add the commercial companies for all topical drugs used.
The quantification of immune cells and TUNEL cell were performed by one researcher? Have authors used a doble blind system?
In the result section:
In page 6 line 211. In the text authors says 79,6% of macrophages that express C3aR, but in the figure the number is 79,5, please correct it.
In the page 6 line 215 authors says in the text “Analysis of retinal cross sections of C3aR KO demonstrated increased TUNEL reactivity in the ONL compared to WT” but in the figure the numbers of TUNEL cell are higher in WT than in C3aR KO. Please correct this.
Author Response
We sincerely thank the reviewer for the valuable comments for the review of our manuscript “Retinal injury activates complement expression in Müller cells leading to neuroinflammation and photoreceptor cell death”. Please see below our point-by-point response addressing the concerns. We have also included a manuscript containing highlighted changes (major edits) as well as a cleaned version of the manuscript. We hope that you agree that it is now in a form appropriate for publication in ‘Cells’.
Reviewer 2
The paper entitle “Retinal injury activates complement expression in Müller cells leading to neuroinflammation and photoreceptor cell death” presented by Tabor SJ et al. is well-structured and results are very interesting for ophthalmology field. However this reviewer has some minor concerns that need to be address before publication.
The introduction although being brief is clear for people who has wealth of knowledge in the field, but if the authors wants to achieve more public they should explain briefly the main photoreceptor death pathways and the relevance of the complement in each one of them. Although this is not mandatory, it is only a suggestion.
We appreciate the reviewer for the comments that our findings are interesting for the ophthalmology field. As suggested by the reviewer, in the introduction we have briefly explained the photoreceptor cell death pathways and the relevance of the complement activation.
In the methods section:
Authors should add the license number of the animal care committee
We have included the approved protocol numbers in the methods section as suggested by the reviewer.
Add the commercial companies for all topical drugs used.
We have included the commercial companies’ names for all topical drugs used in the methods section.
The quantification of immune cells and TUNEL cell were performed by one researcher? Have authors used a doble blind system?
Yes, the samples were masked and given to the investigator for staining and quantification of immune cells and TUNEL positive cells. The results were consistent with the previous findings [1,2].
In the result section:
In page 6 line 211. In the text authors says 79,6% of macrophages that express C3aR, but in the figure the number is 79,5, please correct it.
We apologize for the error, which is rectified now.
In the page 6 line 215 authors says in the text “Analysis of retinal cross sections of C3aR KO demonstrated increased TUNEL reactivity in the ONL compared to WT” but in the figure the numbers of TUNEL cell are higher in WT than in C3aR KO. Please correct this.
Thank you for pointing out the error, we have rectified it now.
References:
- Okunuki, Y.; Mukai, R.; Pearsall, E.A.; Klokman, G.; Husain, D.; Park, D.H.; Korobkina, E.; Weiner, H.L.; Butovsky, O.; Ksander, B.R.; et al. Microglia inhibit photoreceptor cell death and regulate immune cell infiltration in response to retinal detachment. Proc Natl Acad Sci U S A 2018, 115, E6264-E6273, doi:10.1073/pnas.1719601115.
- Sweigard, J.H.; Matsumoto, H.; Smith, K.E.; Kim, L.A.; Paschalis, E.I.; Okonuki, Y.; Castillejos, A.; Kataoka, K.; Hasegawa, E.; Yanai, R.; et al. Inhibition of the alternative complement pathway preserves photoreceptors after retinal injury. Sci Transl Med 2015, 7, 297ra116, doi:10.1126/scitranslmed.aab1482.

Reviewer 3 Report
Retinal detachment (RD) uaually causes photoreceptor cell death, inflammation, and finally vision loss.The complement is consdered to be involved in the pathogenesis of RD, however the source of complement production remains unclarified. In present study, the author showed that the C3 released from the Muller cells activated microglia and caused infiltration of macrophages, which caused the apoptosis of photoreceptors. The finding adds something new in this fields. I have the following major points.
1. The time point of the experiment needs to be extended. Only 24 hours after RD was selected in present experiment. The activation of microglia and the infilitration of macrophage or monocytes usually need more time. The degeneration of photoreceptors usually needs more than one week and the thickness of the out nuclear layer will change. I suggest observe the changes of C3 level in muller cells and the activation of microglia 3 days, one week, two weeks after RD.
2. More cell markers of microglia are required to classify the subpopulation of microglia as they produce quite different effects on the pgotoreceptors.
3. Is any monocyte pass by the blood -retinal barrier? In other retinal degeneration model, the monocyte instead of macrophage was observed in the subretinal space .
4. If the activated microglia was cleared, does the apoptosis of the photoreceptors will be resuced? I suggest the author add more data to support the conclusion.
5. The roles of microglia in the injuried retina has been reported previously, I suggest the author analyze the backgroud in detail.
The quality of the English is OK.
Author Response
We sincerely thank the reviewer for the valuable comments for the review of our manuscript “Retinal injury activates complement expression in Müller cells leading to neuroinflammation and photoreceptor cell death”. Please see below our point-by-point response addressing the concerns. We have also included a manuscript containing highlighted changes (major edits) as well as a cleaned version of the manuscript. We hope that you agree that it is now in a form appropriate for publication in ‘Cells’.
Reviewer 3
Retinal detachment (RD) uaually causes photoreceptor cell death, inflammation, and finally vision loss.The complement is consdered to be involved in the pathogenesis of RD, however the source of complement production remains unclarified. In present study, the author showed that the C3 released from the Muller cells activated microglia and caused infiltration of macrophages, which caused the apoptosis of photoreceptors. The finding adds something new in this fields. I have the following major points.
- The time point of the experiment needs to be extended. Only 24 hours after RD was selected in present experiment. The activation of microglia and the infilitration of macrophage or monocytes usually need more time. The degeneration of photoreceptors usually needs more than one week and the thickness of the out nuclear layer will change. I suggest observe the changes of C3 level in muller cells and the activation of microglia 3 days, one week, two weeks after RD.
We utilized the well-established acute retinal detachment model using sodium hyaluronate. In this model, photoreceptor cell death, immune cell infiltration, and complement activation peaks at 24 hr (shown below as figure 2) (2-4). Per the reviewer’s suggestion, we have extended the time-point and examined C3 expression at day 1 and 3 (please see below Revised Figure 1A included in the manuscript). We also would like to point out that in the sodium hyaluronate induced RD model, by one week, the retina severely degenerates.
The purpose of our study is to determine the cellular source for complement activation at the early/acute phase (24 hrs) of the injury and investigate if the loss of complement components and their cognate receptors mitigates photoreceptor cell death and suppresses infiltration of immune cells into the sub-retinal phase.
We postulate that the neuroprotection of photoreceptors and decreased incidences of recruited immune cells during the acute phase of the injury by targeting complement activation could potentially prolong cell survival before necessary reattachment surgery.
- More cell markers of microglia are required to classify the subpopulation of microglia as they produce quite different effects on the pgotoreceptors.
We agree with the reviewer that subpopulations of microglia have distinct function in the retina. In several neurodegenerative models, including retinal detachment, signature microglial markers are downregulated. It is technically challenging to distinguish the subpopulation by flow cytometry and immunostaining. We therefore used P2ry12, CD11b, and CD45 to distinguish microglia by flow cytometry. We plan to identify the microglial subpopulation by single cell RNA-seq based approach in future studies.
- Is any monocyte pass by the blood -retinal barrier? In other retinal degeneration model, the monocyte instead of macrophage was observed in the subretinal space.
In this acute injury model, immune cells pass the blood-retinal barrier (4), and it is possible that monocytes also infiltrate.
- If the activated microglia was cleared, does the apoptosis of the photoreceptors will be resuced? I suggest the author add more data to support the conclusion.
Our group have already demonstrated that the removal of microglia increases photoreceptor cell death in retinal detachment (4), suggesting neuroprotective functions of microglia. Based on our findings, we postulate that it is not ideal to target microglia directly, rather inhibition of complement activation could potentially suppress peripheral immune cell infiltration and photoreceptor loss.
- The roles of microglia in the injuried retina has been reported previously, I suggest the author analyze the backgroud in detail.
Yes, we agree with the reviewer that including our previous study characterized the role for microglia in retinal injury (2, 4, 5). This study focused on characterizing the cellular source for complement activation and its role in photoreceptor cell death and infiltration of immune cells into the sub-retinal space.
References:
- M. R. Verardo et al., Abnormal reactivity of muller cells after retinal detachment in mice deficient in GFAP and vimentin. Invest Ophthalmol Vis Sci 49, 3659-3665 (2008).
- H. Matsumoto et al., Strain difference in photoreceptor cell death after retinal detachment in mice. Invest Ophthalmol Vis Sci 55, 4165-4174 (2014).
- J. H. Sweigard et al., Inhibition of the alternative complement pathway preserves photoreceptors after retinal injury. Sci Transl Med 7, 297ra116 (2015).
- Y. Okunuki et al., Microglia inhibit photoreceptor cell death and regulate immune cell infiltration in response to retinal detachment. Proc Natl Acad Sci U S A 115, E6264-E6273 (2018).
- S. H. Lee, Y. S. Park, S. S. Paik, I. B. Kim, Differential Response of Muller Cells and Microglia in a Mouse Retinal Detachment Model and Its Implications in Detached and Non-Detached Regions. Cells 10 (2021).

Round 2
Reviewer 3 Report
The manuscript has revised according to the comments of the reviewers and the additonal data were added to support the conclusion. The background and the discussion were revised too.